# Food-washing monkeys recognize the law of diminishing returns

Jessica E Rosien[1,2], Luke D Fannin[1,3], Justin D Yeakel[4,5], Suchinda Malaivijitnond[6,7], Nathaniel J Dominy[1,2]*, Amanda Tan[8]*

[1]Department of Anthropology, Dartmouth College, Hanover, United States; [2]Department of Biological Sciences, Dartmouth College, Hanover, United States; [3]Ecology, Evolution, Environment & Society, Dartmouth College, Hanover, United States; [4]Department of Life and Environmental Sciences, University of California, Merced, Merced, United States; [5]The Santa Fe Institute, Santa Fe, United States; [6]Department of Biology, Chulalongkorn University, Bangkok, Thailand; [7]National Primate Research Center of Thailand, Chulalongkorn University, Saraburi, Thailand; [8]Department of Anthropology, Durham University, Durham, United Kingdom

## eLife Assessment

This is a **valuable** study that tests the functional role of food-washing behavior in removing tooth-damaging sand and grit in long-tailed macaques and whether dominance rank predicts level of investment in the behavior. The evidence that food-washing is deliberate is **compelling** and the evidence that individual investment in the behavior varies is **solid**. Overall, the article should be of interest to researchers interested in foraging behavior, cognition, and primate evolution.

*For correspondence:
nathaniel.j.dominy@dartmouth.edu (NJD);
amanda.tan@durham.ac.uk (AT)

## Abstract

Few animals have the cognitive faculties or prehensile abilities needed to eliminate tooth-damaging grit from food surfaces. Some populations of monkeys wash sand from foods when standing water is readily accessible, but this propensity varies within groups for reasons unknown. Spontaneous food-washing emerged recently in a group of long-tailed macaques (*Macaca fascicularis*) inhabiting Koram Island, Thailand, and it motivated us to explore the factors that drive individual variability. We measured the mineral and physical properties of contaminant sands and conducted a field experiment, eliciting 1282 food-handling bouts by 42 monkeys. Our results verify two long-standing presumptions: monkeys have a strong aversion to sand and removing it is intentional. Reinforcing this result, we found that monkeys clean foods beyond the point of diminishing returns, a suboptimal behavior that varied with social rank. Dominant monkeys abstained from washing, a choice consistent with the impulses of dominant monkeys elsewhere: to prioritize rapid food intake and greater reproductive fitness over the long-term benefits of prolonging tooth function.

## Introduction

Koshima Island, Japan, is a storied fieldsite in the annals of primatology. Observations of wild macaques (*Macaca fuscata*) began in 1948, but 4 years of sustained effort failed to habituate the monkeys. In August 1952, *Itani and Tokuda, 1954* resorted to scattering wheat grains and sweet potatoes along oft-used paths, gradually shifting the provisions to a sandy beach at Otomari Bay, a strategy that afforded a clear view of the entire group of 22 animals. Individual identifications followed quickly, laying the foundation for decades of influential research. In September 1953, a young female named Imo gathered a sweet potato from the beach and rinsed it in a freshwater stream, a behavioral

**eLife digest** While picnics at the beach sound fun, sand is notorious for sticking to food surfaces and nobody likes the feeling of grit on their teeth. Therefore, it is hardly surprising that monkeys living near beaches tend to clean sand from their food. While some briefly brush food with their hands, others wash food items in the ocean with care.

Although washing food in water might seem like the best way to eliminate sand, the most dominant monkey in a social group almost never does it. This difference in cleaning behavior raises the possibility that these monkeys make a shrewd calculation in their minds: is eating more quickly worth the risk of tooth damage from sand?

To explore this idea, Rosien et al. studied wild monkeys living on Koram Island in Thailand, which are known to wash their food in water. Measuring the properties of the sand that stuck to the food showed that 78% of it is made up of quartz – a mineral that is known to damage teeth. The researchers then studied how much time the monkeys spent washing or brushing slices of cucumber with different amounts of sand on them. This revealed that monkeys spent more time cleaning food with more sand on it, confirming the idea that they are averse to sand on their food and intentionally remove it.

The experiments also showed that monkeys tend to spend more time cleaning sand from their food than Rosien et al. had predicted to be necessary, indicating they prioritize this careful cleaning over efficient energy intake. However, this was not the case for the most dominant monkeys in the social group. They favored the quicker but less effective method of brushing food with their hands, suggesting that their main priority is immediate feeding, despite the long-term risk of tooth damage from sand.

The findings will be of interest to evolutionary biologists focused on the tradeoffs between foraging behavior and other vital needs, such as growth, reproduction, and ageing. The experiments also suggest that tooth wear can vary among individuals as a result of different cleaning behaviors, rather than just food types, which will be relevant for future studies by paleoanthropologists.

innovation that spread horizontally to peers and then vertically to older kin (*Kawai, 1965*). By 1958, sweet potato washing had become a group-wide trait with a key modification: the monkeys began using seawater instead of standing freshwater, a preference that continues today seven generations later (*Hirata et al., 2001*). These events have since passed into canon as an example of socially transmitted behavior, or culture, among nonhuman primates (*McGrew, 1998*; *Matsuzawa and McGrew, 2008*; *Matsuzawa, 2015*). This legacy is a venerable one, but it overshadows a fundamental question: *why do monkeys wash their food?* (*Sarabian and MacIntosh, 2015*; *Fiore et al., 2020*). Two tacit assumptions—that sand produces an objectionable sensation on teeth, and that it is prudent to minimize tooth damage—are sufficiently intuitive that formal tests are wanting. In consequence, the mineral and physical properties of contaminant sands are unknown, let alone the efficiency of different cleaning behaviors (*Schofield et al., 2018*). Another enigma concerns the preference of some monkeys to brush food with their hands (*Kawai et al., 1992*), a rapid but seemingly inferior means of sand removal. Food-brushing individuals have been characterized as inept (*Kawai, 1965*) or subordinate (*Watanabe, 1994*) in part because the quartz in sand can cause severe tooth damage (*Lucas et al., 2013*; *Towle et al., 2022*). Yet, carrying food to the ocean is expected to incur energetic costs as well as opportunity costs, factors that impelled us to explore the trade-offs of mitigating sand-mediated tooth wear.

## Koram Island, Thailand

The long-tailed macaques (*Macaca fascicularis*) of Koram Island, Thailand, use stone tools to harvest shellfish, a phenomenon that came to light during the surveys of biodiversity damage that followed the Sumatra-Andaman earthquake and tsunami of December 26, 2004 (*Malaivijitnond et al., 2007*; *Gumert and Malaivijitnond, 2012*; *Tan et al., 2015*). The monkeys became a magnet for tourism, and some visitors began supplying the monkeys with market-sourced fruits (cucumbers, melon, pineapple), jettisoning them onto the beach. These events produce food surfaces with considerable concentrations of sand ($\bar{x} = 3.7 \pm 1.3$ mg mm$^{-2}$), which, in turn, elicit food-washing and food-brushing behaviors among the monkeys. To understand the factors driving these reactions, we examined the

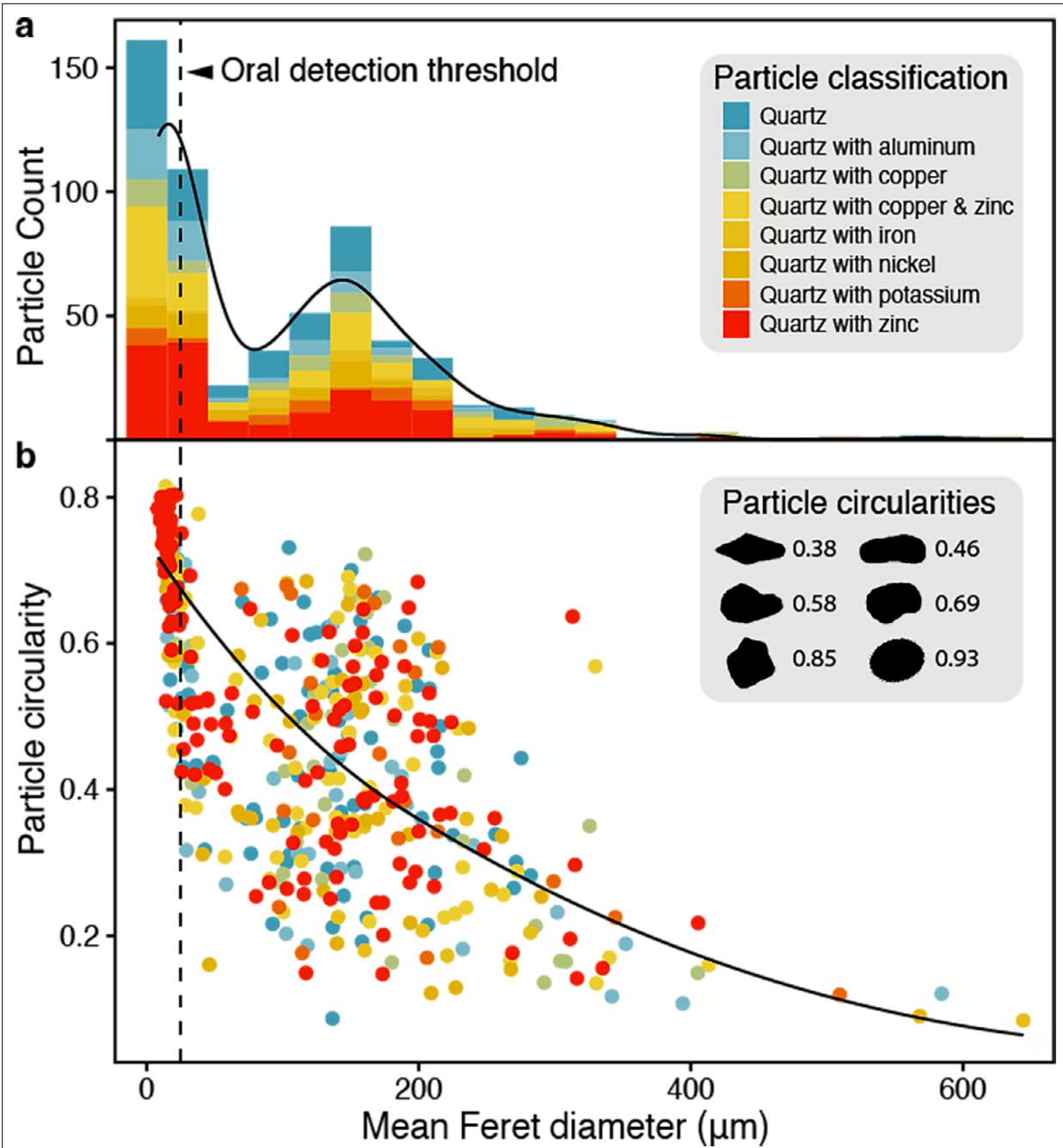

**Figure 1.** Variation in the mineral and physical properties of quartz particles on food surfaces. (**a**) Particle sizes followed a bimodal distribution, with most particles featuring metal inclusions. Nearly half the sample is <25 μm, the 'grittiness threshold' of the human oral cavity (*Imai et al., 1995*). (**b**) Circularity is a dimensionless shape factor (range: 0–1) based on two-dimensional microscopy and estimates for the projected area and perimeter of a given particle. Some examples are illustrated; overall, it is a convenient but imperfect proxy for sphericity, or deviations from spherical (*Grace and Ebneyamini, 2021*). Here, circularity varied as a function of mean Feret diameter, suggesting that larger particles hold greater potential for damaging attack angles during particle–enamel contact.

The online version of this article includes the following figure supplement(s) for figure 1:

**Figure supplement 1.** Comparison of cleaning effectiveness.

mineral properties of food-adhering sands (n = 758 particles), finding that 78% of our sample was composed of crystalline quartz (*Figure 1a*).

Harder than enamel, quartz can exact a heavy toll on teeth, but the probability and degree of enamel loss are governed partly by the size and shape of individual particles, factors that determine the 'attack angle' during particle–enamel contact (*Lucas et al., 2013*). We calculated the circularity of particles as a convenient proxy for sphericity (*Grace and Ebneyamini, 2021*), finding that it decreased as a function of particle size (*Figure 1b*). This result suggests that larger particles are more angular, posing a greater risk to enamel. However, we also calculated a median Feret diameter of 25.8 µm (range: 8.7–644 µm), meaning that nearly half the sample existed below the human threshold (25 µm) of oral detection (*Imai et al., 1995*). To put 25 µm into perspective, it is one-twelfth the diameter of the period ending this sentence.

## Mitigating tooth wear

Chewing undetected quartz is expected to cause severe tooth wear, but this cost can be mitigated behaviorally if food-cleaning is proficient. To test this contention, we simulated the brushing and washing actions of monkeys with cucumber slices exposed to three concentrations of sand: low ($\bar{x}$ = 0.2 ± 0.2 mg mm$^{-2}$), intermediate ($\bar{x}$ = 0.9 ± 0.1 mg mm$^{-2}$), and high ($\bar{x}$ =1.8 ± 0.9 mg mm$^{-2}$). We found that brushing was less efficient than washing across treatments, eliminating 76 ± 7% vs 93 ± 4% of sand particles, respectively (*Figure 1—figure supplement 1*). It is a modest difference, perhaps, but it is freighted with fitness consequences when extrapolated over years of life (*Fannin et al., 2022*). It follows that monkeys should compulsively wash sand from food whenever the opportunity avails itself, a prediction that motivated a field experiment.

## Results

Our experiment was designed to test two concepts at once. The first pivots around intentionality, a thorny problem that emerged from studies of raccoons (*Procyon lotor*). Celebrated food-handlers, the submersion of food objects in water is better termed 'dousing' for greater haptic sensation, not washing with the intention of removing surface contaminants (*Box 1*). If the intent of monkeys is to eliminate sand, then the time devoted to brushing or washing food should vary as a positive function of sandiness. The other concept draws on observations from Koshima Island, which alluded to rank effects on individual cleaning behaviors, a pattern that is difficult to detect without controlling access to food or distance to the ocean.

We conducted 101 feeding trials, recording 1282 food-handling events by 42 individuals (*Figure 2*, *Video 1*). We had detailed rank information for 23 individuals, so we used this subset of data in our generalized linear mixed models (GLMM) analyses. We found that all monkeys were sensitive to sand on their food, responding to each treatment—low, intermediate, and high concentrations—with greater median durations (±1 SD) of brushing (low: 0.0 ± 0.1 s; intermediate: 1.1 ± 2.0 s; high: 3.1 ± 2.0 s; *Figure 2b*) and washing (low: 0.04 ± 0.3 s; intermediate: 0.6 ± 2.0 s; high: 3.3 ± 4.3 s; *Figure 2c*). This result is important for upholding long-held assumptions of intentional cleaning. Further, we found that dominant monkeys of both sexes showed a strong propensity for food-brushing over food-washing (*Figure 2—figure supplement 1*; *Supplementary file 3* and *Supplementary file 4*). This finding reverses the pattern observed on Koshima Island (*Watanabe, 1994*), and it raises the possibility that food-washing is an indulgence subject to diminishing returns. To explore this premise, we developed a theoretical model where the time devoted to food-cleaning is predicted to maximize the rate of sand removal as a function of handling time.

*Figure 3a* illustrates the fastidious nature of our study population: monkeys allocated excess time to washing and brushing—by factors of 1.5 and 3.0, respectively—beyond that predicted by the optimization of sand removal (*Video 2*). Our model also highlights sharply divergent responses to the sunk costs of food-handling time (*Figure 3b*). Given the greater efficacy of washing (*Figure 1—figure supplement 1*) and time needed to carry food to the ocean ($\bar{x}$ = 22 ± 15 s; range: 5–78 s), there is little incentive to over-wash food (*Figure 3b*, region I). At the same time, the lowest- and highest-ranking monkeys abstained from washing altogether, choosing instead to minimize food-handling time by over-brushing their food (*Figure 3b*, region II). This tolerance for fast-diminishing returns underscores

## Box 1. Manual prehension and the nomenclature of raccoon.s

Animal names tend to reflect salient features of their appearance or behavior, including the sounds they make. This basic principle of ethnotaxonomy harmonizes the classification of animals with their traits, a pattern that extends to other languages when names are borrowed or translated. But this process can have profound consequences, shaping our understanding of animals and their behaviors. For example, manipulative actions are baked into the word raccoon, a corruption of the Powhatan words *arakun* or *arakunem*, meaning, approximately, 'it scratches with its hands.' Far more common across North America is the Anishinaabe word *esiban* ('it picks up things'), with cognates in the languages of at least a dozen cultural groups, including Cree, Potawatomi, Abenaki, and Delaware peoples (***Holmgren, 1990***; ***Justice, 2021***). The Tsimshian word *que-o-koo* ('washes with hands') is notably different for imputing intent, perhaps because coastal peoples are sensitized to sand on their foods.

Linnaeus put raccoons in the family Ursidae—bears—in the second edition of *Systema Naturae* (1740), classifying them as *Ursus cauda elongata* ('bear with long tail'). By the 10th edition (1758), he had reclassified them as *Ursus lotor* ('washer bear') due to accounts of captive raccoons persistently dunking food in water, along with his own observations of Sjupp, a pet raccoon gifted to him by the Swedish crown prince Adolf Fredrik (***Nicholls, 2007***). Twenty years later, in 1780, the German naturalist Gottlieb Conrad Christian Storr elevated raccoons into their own genus, *Procyon*, meaning 'before the dog' or 'early dog', a nod to their dog-like appearance; but he retained the species nomen *lotor*. Today, the raccoon is known as *tvättbjörn* in Linnaeus's Swedish and *waschbär* in Storr's German, both meaning 'washer bear.'

Meanwhile, the French naturalist Georges-Louis Leclerc, Comte de Buffon conducted his own detailed observations of pet raccoons held in the Muséum National d'Histoire Naturelle, Paris. He saw affinities with rodents, a legacy that echoes today in the names *raton laveur* and *ratão-lavadeiro* ('washing rat') in French and Portuguese, respectively. So, whether the raccoon is related to bears, dogs, or rats, every European taxonomist of the 18th century agreed that washing was its defining trait. But it is a popular misconception.

Raccoons wet their foods to enhance haptic sensation, not eliminate contaminants; their intent is to douse food objects, not wash them (***Lyall-Watson, 1963***). Intriguingly, the

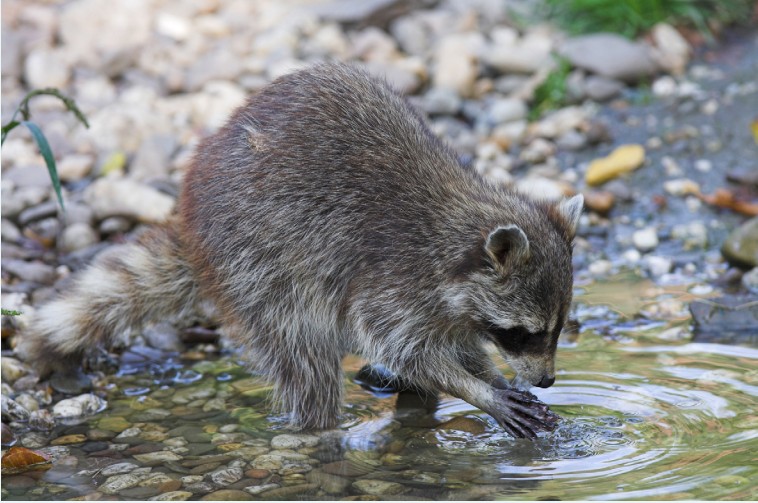

**Box 1—figure 1.** Raccoons (*Procyon lotor*) tend to live in wooded habitats near waterways, where they can be seen dousing foods before ingestion (photograph by Markus Zindl, reproduced with permission).

posterior cerebrum—which includes the somatosensory cortex and several other cortical, thalamic, and subcortical structures—is relatively expanded among raccoons, suggesting magnification of the neural pathways that serve tactile processing (*Arsznov and Sakai, 2013*). When viewed in this light, the Lenape ethnonym *nachenum* ('they use hands as tools') would seem most fitting of all, not least for blurring the thin line between human and nonhuman curiosity and the nature of knowing.

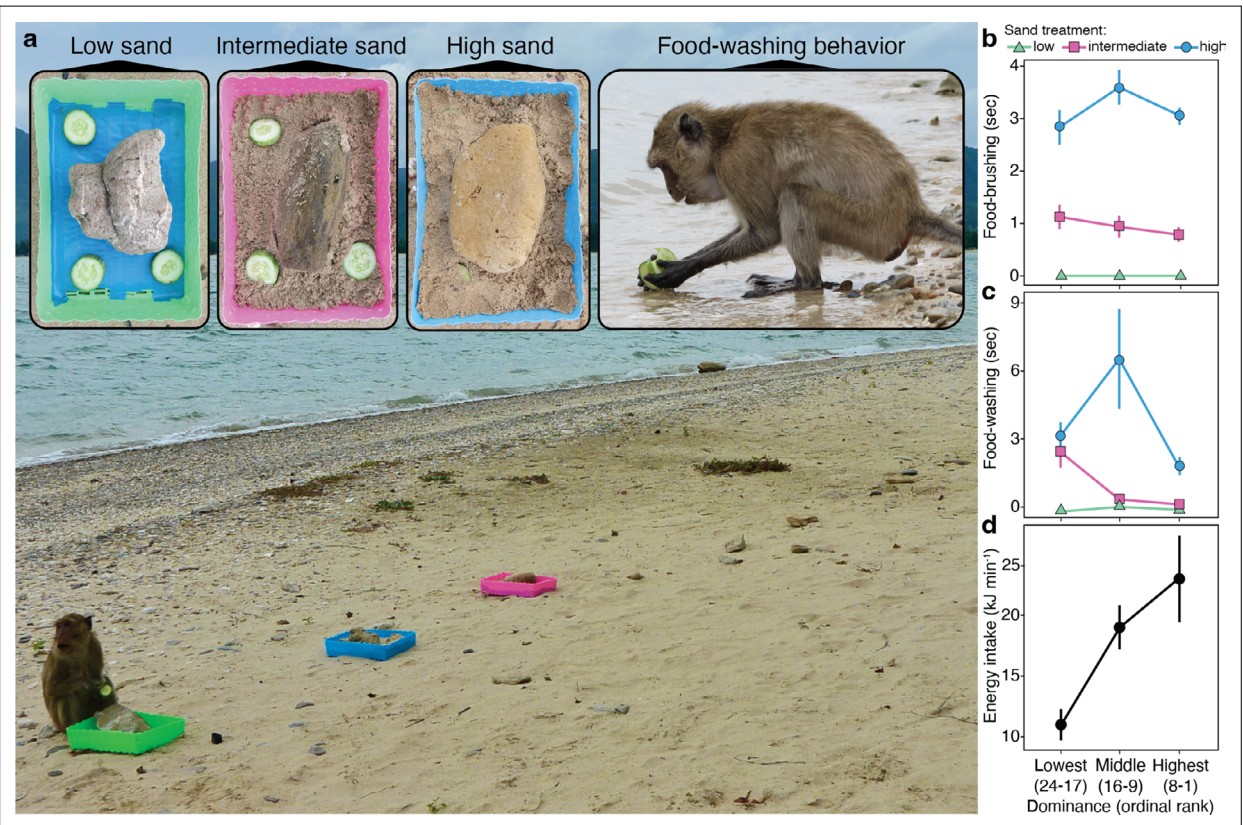

**Figure 2.** Experimental design and results. (**a**) To elicit food-cleaning behaviors, we put sliced cucumbers in trays representing three treatments—food surfaces with low ($\bar{x}$ = 0.2 ± 0.2 mg mm$^{-2}$), intermediate ($\bar{x}$ = 0.9 ± 0.1 mg mm$^{-2}$), and high ($\bar{x}$ = 1.8 ± 0.9 mg mm$^{-2}$) concentrations of sand—positioned 1.5 m apart and 15 m from the ocean. (**b**) Monkeys brushed the sandier treatments for longer durations [$\chi^2$ (2, n = 575 food-handling bouts) = 194.7, p<0.0001] with no effect of dominance rank or sex (*Figure 2—figure supplement 1*). (**c**) Monkeys washed the sandier treatments for longer durations [$\chi^2$ (2, n = 362 food-handling bouts) = 69.7, p<0.0001], and we found an interaction effect with rank independent of sex [$\chi^2$ = 19.3, p<0.0001; *Figure 2—figure supplement 1*]. (**d**) Energy intake rates also varied as function dominance rank [ANOVA, (LN-transformed); $F_{2,104}$ = 10.0; p<0.0001]. Symbols represent mean values and whiskers ± 1 s.e. Photos by Amanda Tan.

The online version of this article includes the following figure supplement(s) for figure 2:

**Figure supplement 1.** Interindividual variation in mean brushing time, washing time, and energy intake.

**Figure supplement 2.** Standardized residual diagnostic plots for the brushing generalized linear mixed models (GLMM) simulated using DHARMa.

**Figure supplement 3.** Standardized residual diagnostic plots for the washing generalized linear mixed models (GLMM), with ordinal rank modeled as a quadratic term, simulated using DHARMa.

**Figure supplement 4.** Standardized residual diagnostic plots for the washing generalized linear mixed models (GLMM), with ordinal rank modeled as a linear term, simulated using DHARMa.

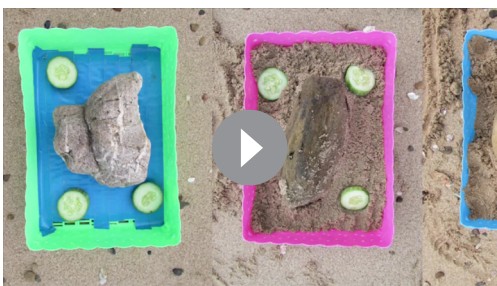

**Video 1.** Experimental design. Video footage of our experimental setting, including our study animals and their cleaning behaviors.
https://elifesciences.org/articles/98520/figures#video1

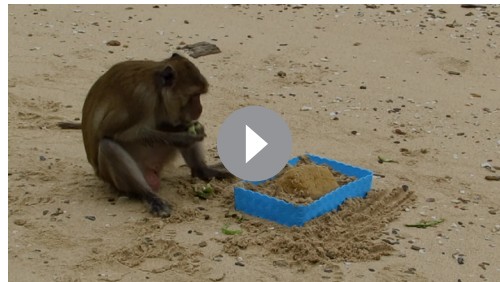

**Video 2.** Examples of overcleaning. Monkeys allocated excess time to washing and brushing—by factors of 1.5 and 3.0, respectively—beyond that predicted by the optimization of sand removal.
https://elifesciences.org/articles/98520/figures#video2

the monkeys' strong aversion to sand; but even so, the long-term benefits of mitigating tooth wear must be balanced against urgent energetic requirements.

## Discussion

Our experiment reveals individual variation at a moment in time. It is a cross-sectional result that can only hint at the long-term fitness consequences of differential cleaning behaviors, a topic with implications for diverse fields of inquiry, from evolutionary theory to paleoanthropology. So, the present discussion was written to put our results into conversation with this wider literature and set an agenda for future research.

### Disposable soma

The disposable-soma hypothesis of senescence predicts investment in the immediate needs of survival or reproduction over tooth preservation (*Carranza et al., 2004*). It is a familiar predicament for

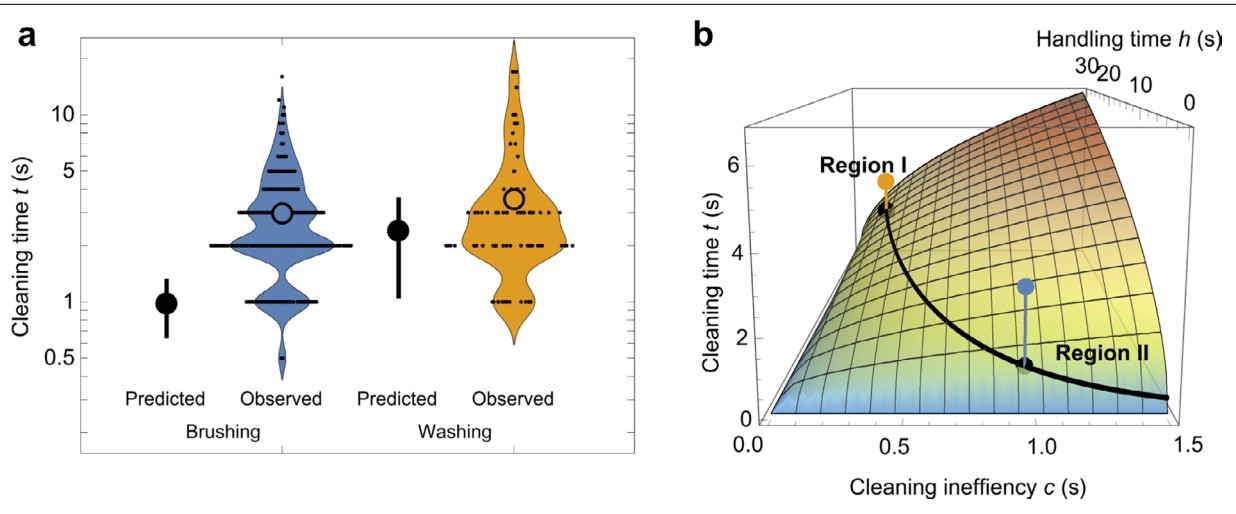

**Figure 3.** Predicted and observed cleaning times. (**a**) Mean predicted time (large filled points vs. observed times [violin plots] for brushing and washing food [note log scale]). The vertical line associated with predicted times represents the 5–95% CI. (**b**) Predicted cleaning time as a function of cleaning inefficiency $c$, and handling time $h$, with mean predicted values (black points) for brushing and washing based on observed cleaning inefficiencies and handling times. The colored points (as in panel **a**) represent observed cleaning times. The trade-off between longer food handling times and efficient cleaning (oceanside food-washing; region I) and shorter handling times and inefficient cleaning (immediate food-brushing; region II) is depicted by the black curve.

The online version of this article includes the following figure supplement(s) for figure 3:

**Figure supplement 1.** Predicted and observed cleaning times.

high-ranking monkeys. In longitudinal studies of macaques and baboons, it is evident that social dominance incurs high energy costs that are met with high feeding rates (*Higham et al., 2011*; *Gesquiere et al., 2025*), behaviors that contribute positively to female fitness (*van Noordwijk and van Schaik, 1999*; *Alberts, 2019*; *Cooper et al., 2022*). Our findings raise the possibility that dominant monkeys refrained from washing to maximize feeding rate (*Figure 2d*). In other words, they prioritized pressing energetic needs over the long-term benefits of tooth preservation—a 'live fast, die young' life-history strategy. This view of teeth as disposable soma could explain why dominant males experience faster senescence and earlier mortality (*Anderson et al., 2021*). Perhaps even a prolonged life is subject to diminishing returns.

## Paleo matter(s)

The full extent of sand-mediated tooth wear is unknown for our study population, but it is probably extreme among the highest-ranked individuals. If affirmed, the findings could affect our views of the hominin fossil record by challenging the assumption that dietary variability is the principal cause of variable dental wear. Some species, notably *Paranthropus boisei*, had ready access to water, which raises the possibility that they—like many primate species—assiduously washed their food, an essential behavior if their diet featured gritty underground plant tissues (*Wrangham et al., 2009*; *Fannin et al., 2021*). Other species, notably *P. robustus*, have extremely variable levels of tooth pitting (*Peterson et al., 2018*), which could reflect, at least partially, the absence of extensive wetlands (*Herries and Hopley, 2010*) coupled with rank effects on food-cleaning behaviors. Tellingly, the dental wear observed on the macaques of Koshima Island bears striking similarities to the hominin fossil record (*Towle et al., 2022*), suggesting that populations of food-cleaning monkeys are a valuable model system that warrant further study.

## Significance

Our study leverages a new method in ecological research to provide the first analysis of siliceous particles on primate foods. Our experiment investigates the behavioral economics of wild monkeys, revealing a strong aversion to sand in their mouths. Yet, the monkeys behaved irrationally when cleaning their foods, allocating excess time than predicted by an optimization model. Some individuals fell victim to the sunk cost fallacy by over-washing their foods (*Box 2*), whereas dominant monkeys abstained from washing altogether, seemingly sacrificing their teeth on the altar of high rank, a social status that depends on rapid food intake. Our results are compatible with the disposable-soma hypothesis for senescence, and they call into question some treasured assumptions in paleoanthropology.

# Materials and methods

## Study site and population

Koram Island (12.242°, 100.009°) lies ~1 km offshore in the Gulf of Thailand and within Khao Sam Roi Yot National Park, Prachuap Khiri Khan, Thailand. It has an area of 0.45 km² and a coastline of 3.5 km. The habitat—limestone karst blanketed with a dense flora of dwarf evergreen trees and deciduous scrub, and encircled by rocky shore and sandy beaches—supports a population of ca. 75 long-tailed macaques described as hybrids at the subspecies taxonomic level (*Macaca fascicularis aurea* × *M. f. fascicularis*) (*Gumert et al., 2019*). The animals are well habituated to human observers due to regular tourism and sustained study since 2013 (*Tan et al., 2018*). Most of this research has revolved around stone tool-mediated foraging on mollusks, the only activity known to elicit stone handling (*Malaivijitnond et al., 2007*; *Gumert and Malaivijitnond, 2012*; *Gumert and Malaivijitnond, 2013*; *Tan et al., 2015*), although infants and juveniles will sometimes use stones during object play (*Tan, 2017*). There has been no prior examination of food-cleaning behaviors.

## Rank determination

Macaques form multi-male multi-female (polygynandrous) social groups with individual dominance hierarchies. In *M. fascicularis*, the hierarchy is strictly linear and extremely steep, meaning aggression is unidirectional (*de Waal, 1977*; *van Noordwijk and van Schaik, 2001*) with profound asymmetries in outcomes for individuals of adjacent ranks (*Balasubramaniam et al., 2012*). Further, the dominance hierarchies of philopatric females are stable and predictable. Daughters follow the pattern

## Box 2. Balancing sunk costs against future benefits.

Carrying food to the ocean costs time and energy. Some monkeys stood upright and walked to the water because their hands held cucumbers, paying an extremely high energetic cost (*Nakatsukasa et al., 2006*). Such irrecoverable expenses—or 'sunk costs'—should not sway the optimal washing time of rational monkeys, which we calculated as 2.40 ± 0.74 s per cucumber slice. Still, many monkeys washed their food far beyond the point of diminishing returns (*Figure 3a*). This level of over-washing speaks to their aversion to sand, but it also suggests that some individuals succumbed to the sunk cost fallacy, or 'Concorde Effect' (*Arkes and Ayton, 1999*).

The Concorde is a supersonic aircraft. Production began in 1962 as a joint Anglo-French enterprise, but significant cost overruns undermined any chance of profit. By 1971, the opening line of a British central policy review staff memorandum was explicit and brutal: "Concorde is a commercial disaster," it said. "It should never have been started." Still, both governments continued to pour millions into the project on the grounds that they had already placed substantial investment in it. Such reasoning is viewed as an economic fallacy because decisions should be based on prospective (future) costs only. The same memo put it this way, "The decision whether or not to abandon Concorde must start from where we are now—much of the milk is already spilt."

Irrational decisions are not unique to humans. When faced with a foraging task that required inaction, humans, rats, and mice each fell victim to the Concorde Effect; they were more likely to complete a trial (i.e., to continue waiting instead of opting out) the longer they had already waited (*Sweis et al., 2018*). In another experiment, *Watzek and Brosnan, 2020* showed that tufted capuchins and rhesus macaques are reluctant to forfeit a small investment when completing a psychomotor task, persisting 5–7 times longer than was optimal for a food reward. A criticism of these findings is that animal hunger can bias the results, impelling irrational decisions (*Ott et al., 2022*). In our experiment, however, middle-ranking monkeys delayed consumption of a food reward already in their possession, suggesting an incentive to offset the long-term costs of sand-mediated tooth damage. If irrational over-washing confers adaptive benefits by prolonging the functional life of teeth, then there could be a kernel wisdom in the Concorde Effect.

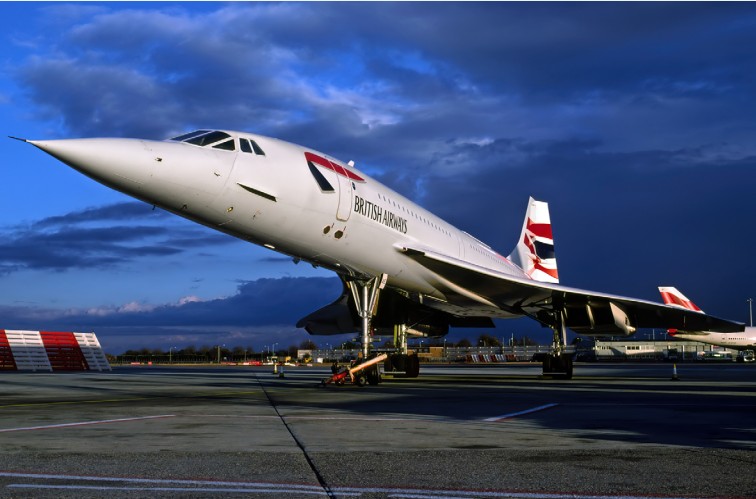

**Box 2—figure 1.** The Aérospatiale-BAC Concorde 102 is an iconic aircraft manufactured from 1965 to 1979. This photo shows British Airways flight 002 on the eve of its final commercial flight on October 24, 2003 (photograph by Richard Vandervord, reproduced with permission).

of youngest ascendancy, ranking just below their mothers with few known exceptions among older sisters (*de Waal, 1977*; *van Noordwijk and van Schaik, 1999*). Taken together, these species traits are conducive to unequivocal rank determinations.

To determine the rank order of adults in our study group, we recorded dyadic agonistic interactions and their outcomes (i.e., aggression, supplants, and silent-bared-teeth displays of submission) during 5 min focal follows of individuals based on a randomized order of continuous rotation (*Tan et al., 2018*). These data were supplemented with ad libitum observations and all rank determinations were updated monthly, and when males immigrated or emigrated. This protocol predates our experiment in July–August 2018, representing ~970 hr of focal data during five years of systematic study (2013–2018).

To determine the effects of dominance rank on individual food-cleaning propensities, we followed the methods of *Levy et al., 2020*. We calculated ordinal ranks between 1 (highest) and n (lowest), where n is the number of animals aged ≥5 years. Then we combined males (n = 8) and females (n = 16) into a single standardized ordinal ranking of 24 animals. This method of rank determination is well suited for conditions involving density-dependent competition (*Levy et al., 2020*), such as those of our experiment and more generally on Koram Island, where preferred food resources are limited (*Luncz et al., 2017*).

## Measuring sand

To quantify the amount of sand on food surfaces, whether provisioned by tourists (cucumbers, melon, pineapple) or used in our experiment (sliced cucumbers), we applied a quick-drying liquid polymer—granulated plastic (Pioloform BL 16; Wacker-Chemie GMBH, Munich, Germany) mixed with ethanol (18% plastic to 82% ethanol, by weight)—to each food item. When dried, we peeled and stored the sand-infused film for analysis. The advantages of this method are twofold: the removal of exogenous particulate matter is extremely thorough; and the plastic does not detach biogenic silica such as trichromes or phytoliths (*Hinton et al., 1996*). In the lab, we dissolved each peel in ethanol and separated the sand by centrifugation, producing a pellet. We dried and weighed the pellet, dividing the mass by the surface area of the food object, which we calculated from digital photographs imported into ImageJ v. 1.52. This method produces an estimate of exogenous particulate mass per area (mg mm$^{-2}$), allowing direct comparison of apples and oranges.

To measure the elemental and physical properties of sand, we dispersed and filtered the pellets in water using a 0.2 µm isopore membrane filter, which we submitted for scanning electron microscopy and energy-dispersive X-ray spectroscopy. To establish the parameters for multi-field and bulk analysis, we imaged a representative area of the filter at multiple magnifications and performed discrete particle analysis. 50× magnification allowed for a statistically significant representation of particle number and size range (allowing a 5 µm lower particle size range in analysis). All discrete particle analyses indicated silicon-rich particles, and composition distribution bins were established to include dominant accompanying elements. After establishing these parameters, we initiated multi-field automated analysis using six fields of view of the debris field (at 50×). A composition classification was assigned to each particle and data sorted by composition classification and particle size (particle sizing was binned using standard Feret maximum parameter). The sizing bins are standard ISO-16232 size classes.

## Experimental design

To elicit food-handling behaviors and determine individual cleaning preferences, we put three cucumber slices in each of three trays (20 × 30 × 10 cm) and manipulated the amount of contaminating sand. In the low-sand treatment, we put cucumber slices in a tray without sand; however, contamination from aeolian sand was unavoidable. In the intermediate-sand treatment, we lined the tray with sand and put the cucumber slices on the surface. In the high-sand treatment, we buried cucumber slices completely (*Figure 2a*). Trays were placed 1.5 m apart and 15 m from the ocean, and we randomized the color and sequence of trays across trials.

Trials began when one or more monkeys approached the trays and ended when the animals finished every cucumber slice or abandoned the experiment (range: 10 s to 14 min). We used video recordings to determine the onset and offset of individual food-handling bouts, beginning from initial contact with a cucumber slice and ending when the final slice, in its entirety, entered the mouth. Within each

bout, we determined the duration of brushing and washing behaviors, defining each from the onset of serial stereotypical forelimb movements to the moment of oral ingestion. We estimated energy intake rates by calculating the number of cucumber slices consumed during each food-handling bout and multiplying each slice by 1.1 kcal (source: U.S. Department of Agriculture, FoodData Central, 2019) and 4.186 kJ (*Hargrove, 2007*). We performed 101 trials over 5 weeks and recorded 1282 food-handling bouts by 42 individual monkeys. We excluded trials from analysis if the number of participating monkeys exceeded the number of feeding stations as these conditions produced high levels of feeding competition with scant cleaning behavior. Such conditions effectively erased individual variation in sand removal, the topic motivating our experiment. Accordingly, we analyzed trials with ≤3 monkeys, putting 937 food-handling bouts into the GLMM statistical models, which included data on individual rank, sex, and sand treatment. If a monkey consumed a cucumber slice without brushing or washing it, the zero-second duration was included in both GLMMs.

## Behavioral analyses

To model variance in brushing and washing behaviors as a function of experimental treatment and rank, we fit GLMMs using the *glmmTMB* package in R version 4.4.1 (*Brooks et al., 2017*). We then evaluated the performance of each model using the standardized residual diagnostic tools for hierarchical regression models available in the DHARMa package (*Hartig, 2022*). In each GLMM, we modeled *sand treatment* (a categorical variable with three levels), *sex* (a categorical variable with two levels), and *ordinal rank* (a discrete variable ranging from 1 to 24) as fixed effects. For our brushing model, we incorporated an additional interaction term as a fixed effect: *sand treatment* × *ordinal* rank. For our washing dataset, we ran two models: the first modeled the relationship between ordinal rank and food cleaning as a linear term, while the second modeled the relationship as a quadratic term. We also incorporated both the linear and quadratic terms into the interaction effect with *sand treatment* in each washing model. To account for experimental variance among individuals and control for pseudoreplication (because the number of feeding bouts per individual varied widely; *Bolker et al., 2009*), we included individual ID as a random intercept.

Our brushing and washing datasets were whole-number counts (seconds) with means < 5. The distributions were right-skewed with high concentrations of biologically meaningful zeros (*Martin et al., 2005*) (i.e., instances of food-handling without any cleaning behavior). Thus, we fit a series of zero-inflated generalized linear mixed models (ZIGLMM), each with a logit-link function, a single zero-inflation parameter applying to all observations, and Poisson error distribution. For the food-brushing model, we fit a zero-inflated Poisson (ZIP), which produced favorable standardized residual diagnostic plots with no major patterns of deviation (*Figure 2—figure supplement 2*) and minor, but nonsignificant, underdispersion (DHARMa dispersion statistic = 0.99, p=0.80). For our two food-washing models, we used zero-inflated models with Conway–Maxwell Poisson (ZICMP) distributions, an error distribution chosen for its ability to handle data that is more underdispersed (DHARMa dispersion statistic = 8.2E-09, p=0.74) than the standard zero-inflated Poisson (*Brooks et al., 2019*). Using this error distribution improved residual diagnostic plots over a standard ZIP model, and we view any deviations in the standardized residuals as minor and due to the smaller sample size of our food-washing dataset (*Figure 2—figure supplements 3 and 4*; *Hartig, 2022*). We report the summarized fixed-effects tests for each GLMM in *Figure 2—figure supplements 1–3* as Analysis of Deviance Tables (type II Wald $\chi^2$ tests, one-sided) along with $\chi^2$ values, degrees of freedom, and p-values (one-sided tests). Full model summaries with SEs and CIs are also included in *Supplementary files 4–6*. For all statistical analyses, we set α = 0.05.

## Optimal cleaning time model

To model the optimization of sand removal, we drew inspiration from the marginal value theorem of *Charnov, 1976*, defining two temporal periods: handling time $h$, which includes an assessment time and pre-cleaning time, and cleaning time $t$. Assessment time (set as a constant 1 s) includes visual fixation on a food object and forelimb extension before contact, whereas pre-cleaning time represents all handling activities that precede cleaning. During brushing, the pre-cleaning time was essentially nil (zero seconds), but washing required travel from the experimental treatments to the ocean, requiring longer pre-cleaning times ($\bar{x} = 22 \pm 15$ s; range: 5–78 s). We assumed that the proportion of sand removed from each cucumber follows the saturating relationship $g(t) = t/(c + t)$, where $c$ is the cleaning

inefficiency, or the half-saturation constant associated with brushing or washing. As $c$ increases, so does the inefficiency of a given cleaning behavior. Given our observations, it requires an average of 2.97 s of brushing to remove 75 ± 7% of grit, and 3.53 s of washing to remove 93 ± 4% of grit (*Figure 1—figure supplement 1*). From these data, and including the experimental uncertainty associated with grit removal percentages, we obtained distributions for estimated cleaning inefficiencies, $c_{\text{brushing}} = 0.99 \pm 0.38$ s and $c_{\text{washing}} = 0.27 \pm 0.15$ s, such that washing (without considering handling costs) is the most efficient strategy. The rate of grit removal is then given by $R(t) = g(t)/(h + t)$, which reaches a maximum at the optimal cleaning time $t^* = \sqrt{ch}$. For brushing and washing cleaning strategies, we obtain the expected optimal cleaning times $t^*_{\text{brushing}} = 0.98 \pm 0.19$ s, and $t^*_{\text{washing}} = 2.40 \pm 0.74$ s (*Figure 3a*), respectively, and where including additional sources of uncertainty did not alter our findings (*Figure 3—figure supplement 1*). These optimal cleaning times are defined exclusively to maximize the rate of grit removal, without considering the potentially cascading effects of these strategies on fitness.

## Acknowledgements

We are extremely grateful for the guidance and practical assistance of G Badihi, C Hobaiter, J Hua, L Kaufman, WC McGrew, A Mielke, D Pornsumrit, and ZM Thayer. This research was approved by the Institutional Animal Care and Use Committee of Dartmouth College (protocol no. 00002099), the National Research Council of Thailand (permit nos. 0002/3740 and 0002/3742), and the Department of National Parks, Wildlife and Plant Conservation of Thailand. Funding was received from the National Science Foundation (BCS-SBE 1829315 to NJD; GRFP 1840344 to LDF) and Dartmouth College, including awards to JER (Claire Garber Goodman Fund; Mark A Hansen Undergraduate Research, Scholarship, and Creativity Fund; Student Experiential Learning Fund) and NJD (Scholarly Innovation and Advancement Award).

## Additional information

### Competing interests

Justin D Yeakel: Reviewing editor, eLife. The other authors declare that no competing interests exist.

### Funding

| Funder | Grant reference number | Author |
|---|---|---|
| National Science Foundation | BCS-SBE 1829315 | Nathaniel J Dominy |
| National Science Foundation | Graduate Research Fellowship Program 1840344 | Luke D Fannin |
| Dartmouth College | Claire Garber Goodman Fund | Jessica E Rosien |
| Dartmouth College | Mark A. Hansen Undergraduate Research, Scholarship, and Creativity Fund | Jessica E Rosien |
| Dartmouth College | Student Experiential Learning Fund | Jessica E Rosien |
| Dartmouth College | Scholarly Innovation and Advancement Award | Nathaniel J Dominy |

The funders had no role in study design, data collection and interpretation, or the decision to submit the work for publication.

### Author contributions

Jessica E Rosien, Conceptualization, Data curation, Funding acquisition, Investigation, Methodology, Writing – review and editing; Luke D Fannin, Data curation, Formal analysis, Visualization,

Methodology, Writing – original draft, Writing – review and editing; Justin D Yeakel, Formal analysis, Visualization, Methodology, Writing – original draft, Writing – review and editing; Suchinda Malaivijitnond, Investigation, Methodology, Project administration, Writing – review and editing; Nathaniel J Dominy, Formal analysis, Supervision, Funding acquisition, Writing – original draft, Project administration, Writing – review and editing; Amanda Tan, Conceptualization, Supervision, Investigation, Methodology, Writing – original draft, Writing – review and editing

### Author ORCIDs
Jessica E Rosien ⓘ https://orcid.org/0000-0001-9991-2138
Luke D Fannin ⓘ https://orcid.org/0000-0003-4810-4442
Justin D Yeakel ⓘ http://orcid.org/0000-0002-6597-3511
Suchinda Malaivijitnond ⓘ https://orcid.org/0000-0003-0897-2632
Nathaniel J Dominy ⓘ https://orcid.org/0000-0001-5916-418X
Amanda Tan ⓘ https://orcid.org/0000-0002-9227-4644

### Ethics
This research was approved by the Institutional Animal Care and Use Committee of Dartmouth College (protocol no. 00002099), the National Research Council of Thailand (permit nos. 0002/3740 and 0002/3742), and the Department of National Parks, Wildlife and Plant Conservation of Thailand.

Reviewer #1 (Public review): https://doi.org/10.7554/eLife.98520.4.sa1
Author response https://doi.org/10.7554/eLife.98520.4.sa2

## Additional files

### Supplementary files
Supplementary file 1. Summarized fixed effects for the food brushing GLMM (n = 575 events by animals with known rank) as an analysis of deviance table (Type II Wald Chi Square Tests).

Supplementary file 2. Summarized fixed effects for the food washing GLMM (n = 362 events by animals with known rank) as an analysis of deviance table (Type II Wald Chi Square Tests) for the model that included both quadratic and linear ordinal rank terms.

Supplementary file 3. Summarized fixed effects for the food washing GLMM (n = 362 events by animals with known rank) as an analysis of deviance table (Type II Wald Chi Square Tests) for the model that included just a linear ordinal rank term.

Supplementary file 4. Full model fixed effects and confidence intervals for the food brushing GLMM. Modeled as a zero-inflated Poisson (ZIP) (n = 575 observations).

Supplementary file 5. Full model fixed effects and confidence intervals for the food washing GLMM fixed effects with both a quadratic and linear ordinal rank term. Modeled as a zero-inflated Conway-Maxwell Poisson (ZICMP) (n = 362 observations).

Supplementary file 6. Full model fixed effects and confidence intervals for the food washing GLMM fixed effects with only a linear ordinal rank term. Modeled as a zero-inflated Conway-Maxwell Poisson (ZICMP) (n = 362 observations).

MDAR checklist

### Data availability
All project data and code are available in the Zenodo repository. The data and code for producing *Figure 3* are contained in a Mathematica notebook (v. 14.0), also available in the Zenodo repository: https://doi.org/10.5281/zenodo.10513540.

The following dataset was generated:

| Author(s) | Year | Dataset title | Dataset URL | Database and Identifier |
| --- | --- | --- | --- | --- |
| Rosien JE, Fannin LD, Yeakel JD, Malaivijitnond S, Dominy NJ, Tan A | 2024 | Data for: Food-washing monkeys recognize the law of diminishing returns | https://doi.org/10.5281/zenodo.10513540 | Zenodo, 10.5281/zenodo.14002737 |

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
