## [Editor Report · eLife Assessment]

This is a **valuable** study that tests the functional role of food-washing behavior in removing tooth-damaging sand and grit in long-tailed macaques and whether dominance rank predicts level of investment in the behavior. The evidence that food-washing is deliberate is **compelling** and the evidence that individual investment in the behavior varies is **solid**. Overall, the article should be of interest to researchers interested in foraging behavior, cognition, and primate evolution.

---

## [Referee Report · Reviewer #1 (Public review)]

In this paper, the authors had 2 aims:

(1) Measure macaques' aversion to sand and see if its' removal is intentional, as it likely in an unpleasurable sensation that causes tooth damage.

(2) Show that or see if monkeys engage in suboptimal behavior by cleaning foods beyond the point of diminishing returns, and see if this was related to individual traits such as sex and rank, and behavioral technique.

They attempted to achieve these aims through a combination of geochemical analysis of sand, field experiments, and comparing predictions to an analytical model.

The authors' conclusions were that they verified a long-standing assumption that monkeys have an aversion to sand as it contains many potentially damaging fine grained silicates, and that removing it via brushing or washing is intentional.

They also concluded that monkeys will clean food for longer than is necessary, i.e. beyond the point of diminishing returns, and that this is rank-dependent.

High and low-ranking monkeys tended not to wash their food, but instead over-brushed it, potentially to minimize handling time and maximize caloric intake, despite the long-term cumulative costs of sand.

This was interpreted through the *disposable soma hypothesis*, where dominants maximize immediate needs to maintain rank and increase reproductive success at the potential expense of long-term health and survival.

Strengths:

The field experiment seemed well designed, and their quantification of the physical and mineral properties of quartz particles (relative to human detection thresholds) seemed good relative to their feret diameter and particle circularity (to a reviewer that is not an expert in sand). The *Rank Determination* and *Measuring Sand* sections were clear.

In achieving Aim 1, the authors validated a commonly interpreted, but unmeasured function, of macaque and primate behavior-- a key study/finding in primate food processing and cultural transmission research.

I commend their approach in trying to develop a quantitative model to generate predictions to compare to empirical data for their second aim.

This is something others should strive for.

I really appreciated the historical context of this paper in the introduction and found it very enjoyable and easy to read.

I do think that interpreting these results in the context of the *disposable soma hypothesis* and the potential implications in the *paleolithic matters* section about interpreting dental wear in the fossil record are worthwhile.

---

## [Author Response]

The following is the authors’ response to the previous reviews

We thank the editors and Reviewers 1 and 3 for their though6ul consideration of our manuscript. The present revision is submitted to address comments raised concerning rank determinations and the following sentence in the editorial assessment:

The evidence that food-washing is deliberate is compelling, but the evidence for variable and adaptive investment depending on rank, including the fitness-relevance and ultimate evolutionary implications of the findings, is incomplete given limitations of the experimental design.

Close reading of this sentence reveals two parallel threads. The first can be read as “…evidence for variable rank is incomplete given the limitations of the experimental design,” whereas the second can be read as “…evidence for adaptive investment and fitness is incomplete given the limitations of the experimental design.” The first alludes to a critique of our methods, while the second alludes to points of discussion unrelated to our experimental design. Unpacking this sentence is important because it casts the totality of our paper as “incomplete,” a word of consequence for early-career scholars because it prevents indexing in Web of Science.

For clarity, we will refer to these topics as Thread 1 and Thread 2 in the following response.

Thread 1 seems rooted in a comment made by Reviewer 1, which is reproduced below:

I am still struck that there was an analysis of only trials where <3 individuals are present. If rank was important, I would imagine that behavior might be different in social contexts when theA, scrounging, policing, aggression, or other distractions might occur-- where rank would have effects on foraging behavior. Maybe lower rankers prioritize rapid food intake then. If rank should be related to investment in this behavior, we might expect this to be magnified (or different) in social contexts where it would affect foraging. It might just be that the data was too hard to score or process in those settings, or the analysis was limited. Additionally, I think that more robust metrics of rank from more densely sampled focal follow data would be a beJer measure, but I acknowledge the limitations in getting the ideal. Since rank is central to the interpretation of these results, I think that reduced social contexts in which rank was analyzed and the robustness of the data from which rank was calculated and analyzed are the main weaknesses of the evidence presented in this paper.

We are grateful for this perspective of Reviewer 1, but it puts us in an uncomfortable position. We must respond rather forcefully because of its influence on the above assessment. A problem with R1’s comment is that it uses the word “foraging” (a behavior we did not study) instead of “cleaning” (the behavior we did study). Still, we can substitute the latter word with the former to get the gist of it.

R1 criticizes our methods as a prelude for imagining the behaviors of our study animals, a form of conjecture. R1 correctly supposes a positive relationship between the number of animals and the intensity of competition for a limited food resource, a well-known phenomenon; and, yes, the food in each trial was decidedly limited, being fixed at nine cucumber slices. But R1 incorrectly presumes rank effects on cleaning under conditions of intense food competition. When the number of monkeys participating in a trial exceeded the number of feeding stations (n = 3), we saw little or no cleaning effort, either brushing or washing. So, rank effects on cleaning are immaterial under these conditions. As our study goals were narrowly focused on detecting individual propensities, or choices, as a function of rank, we limited our analysis to trials involving three monkeys or fewer. In retrospect, we admit that we should have provided better justification for our choice of trials, so we’ve edited one of our sentences:

Original sentence

Formerly lines 219-220: To minimize the potential confounding effects of dominance interactions, we analyzed trials with ≤ 3 monkeys.

Revised sentence

Current lines 219-224: We excluded trials from analysis if the number of participating monkeys exceeded the number of feeding stations, as these conditions produced high levels of feeding competition with scant cleaning behavior. Such conditions effectively erased individual variation in sand removal, the topic motivating our experiment. Accordingly, we analyzed trials with ≤ 3 monkeys, putting 937 food-handling bouts into the GLMM statistical models, which included data on individual rank, sex, and sand treatment.

R1’s final criticism – “I think that more robust metrics of rank from more densely sampled focal follow data would be a better measure, but I acknowledge the limitations in getting the ideal” – seems to imply that rank data were collected during our experiment. On the contrary, we determined ranks from five years of focal follows preceding the experiment, achieving the very standard that R1 describes as ideal. The relevant text appeared on lines 165-169 in version 2.0:

To determine the rank-order of adults, we recorded dyadic agonistic interactions and their outcomes (i.e., aggression, supplants, and silent-bared-teeth displays of submission) during 5min focal follows of individuals based on a randomized order of continuous rotation (Tan et al., 2018). In some cases, these data were supplemented with ad libitum observations. This protocol existed during five years (2013-2018) of continual observations before we conducted our experiment in July-August 2018.

Naturally, we were puzzled by R1’s dismissal of our methods, as well as R1’s conclusion, reached without evidence, that “[the] reduced social contexts in which rank was analyzed and the robustness of the data from which rank was calculated and analyzed are the main weaknesses of the evidence presented in this paper.” It is unsubstantiated assertation with no definition of robustness, making it difficult for anyone to objectively assess the quality of our data.

We detect in R1’s words some unfamiliarity with the social organization of our study species, which is fair enough. To better orient readers to the dominance hierarchy of Macaca fascicularis, and to boost reader confidence in the volume and quality of our rank data, we have added several sentences to this section of the manuscript, lines 169-183:

Macaques form multi-male multi-female (polygynandrous) social groups with individual dominance hierarchies. In M. fascicularis, the hierarchy is strictly linear and extremely steep, meaning aggression is unidirectional (de Waal, 1977; van Noordwijk and van Schaik, 2001) with profound asymmetries in outcomes for individuals of adjacent ranks (Balasubramaniam et al., 2012). Further, the dominance hierarchies of philopatric females are stable and predictable. Daughters follow the pattern of youngest ascendancy, ranking just below their mothers with few known exceptions among older sisters (de Waal, 1977; van Noordwijk and van Schaik, 1999). Taken together, these species traits are conducive to unequivocal rank determinations.

To determine the rank-order of adults in our study group, we recorded dyadic agonistic interactions and their outcomes (i.e., aggression, supplants, and silent-bared-teeth displays of submission) during 5-min focal follows of individuals based on a randomized order of continuous rotation (Tan et al., 2018). These data were supplemented with ad libitum observations and all rank determinations were updated monthly, and when males immigrated or emigrated. This protocol predates our experiment in July-August 2018, representing 970 hr of focal data during five years of systematic study (2013-2018).

Thread 2 criticizes our evidence for adaptive investment and fitness, describing it is a limitation of our experimental design. Accordingly, the totality of our experiment was classified as “incomplete.” Yet, our experiment was never designed to collect such evidence, and we make no claims of having it. Rather, we discussed potential fitness consequences to highlight the broader significance of our study, connecting it diverse bodies of literature, from evolutionary theory to paleoanthropology. Our intent was to follow the conventions of scientific writing; to put our results into conversation with the wider literature and set an agenda for future research.

On reflection, Thread 2 seems to pivot around something as arbitrary as structure. Previously, our results and discussion were combined under a single section header (“Results and Discussion”), a stylistic choice to economize words. Our manuscript is a Short Report, which is limited to 1,500 words of main text. But this level of concision proved counterproductive. It blurred our results and discussion in the minds of readers. Indeed, Reviewer 3 described it as “misleading,” a barbed word that accomplishes the same act attributed to us. To counter this perspective, we have simply partitioned our Results (now “Experimental Results”) and Discussion to draw a sharper distinction between the two components of our paper.